# Metabolic Response to Tick-Borne Encephalitis Virus Infection and Bacterial Co-Infections

**DOI:** 10.3390/pathogens11040384

**Published:** 2022-03-23

**Authors:** Marta Dobrzyńska, Anna Moniuszko-Malinowska, Iwona Jarocka-Karpowicz, Piotr Czupryna, Monika Groth, Elżbieta Skrzydlewska

**Affiliations:** 1Department of Analytical Chemistry, Medical University of Białystok, Mickiewicza 2D, 15-222 Białystok, Poland; 34652@student.umb.edu.pl (M.D.); iwona.jarocka-karpowicz@umb.edu.pl (I.J.-K.); 2Department of Infectious Diseases and Neuroinfections, Medical University of Białystok, Żurawia 14, 15-540 Białystok, Poland; anna.moniuszko-malinowska@umb.edu.pl (A.M.-M.); piotr.czupryna@umb.edu.pl (P.C.); monika.groth@umb.edu.pl (M.G.)

**Keywords:** TBEV, infections, bacterial co-infections, lipids modifications, proteins modifications

## Abstract

Ticks are vectors of various pathogens, including tick-borne encephalitis virus and bacteria such as *B. burgdorferi* and *A. phagocytophilum*, causing infections/co-infections, which are still a diagnostic and therapeutic problem. Therefore, the aim of this study was to compare the effects of TBEV infection/bacterial co-infection on metabolic changes in the blood of patients before and after treatment. It was found that those infections promote plasma ROS enhanced generation and antioxidant defence reduction, especially in relation to glutathione and thioredoxin systems, despite the increased effectiveness of Nrf2 transcription factor in granulocytes. Observed oxidative stress promotes the oxidative modifications of phospholipids containing polyunsaturated fatty acids (LA, AA, EPA) with increased lipid peroxidation (estimated as 8-isoPGF2α, 4-HNE). It is accompanied by protein modifications measured as 4-HNE-protein adducts, carbonyl groups, dityrosine increase, and tryptophan level decrease, which promote structural and functional modification of the following transcription factors: Nrf2 and NFkB inhibitors. The lower level of 8-iso-PGF2α in co-infections indicates an impairment of the body’s ability to intensify inflammation and fight co-infections, while an increased level of Trx after therapy may contribute to the intensification of the inflammatory process. The obtained results indicate the potential possibility of using the assessed metabolic parameters to introduce targeted pharmacotherapy in cases of TBEV infections/bacterial co-infections.

## 1. Introduction

Arthropods, including mosquitoes and ticks that live in the human environment, can be vectors of pathogens such as Japanese encephalitis virus (JEV), West Nile virus (WNV), and tick-borne encephalitis virus (TBEV) [1]. Due to the increasing number of ticks in the natural environment of humans, more and more infections with TBEV are observed, which when delivered to the human body causes tick-borne encephalitis (TBE) [2], a disease that continues to be a diagnostic and therapeutic problem.

TBEV belongs to the *Flaviviridae* family and, as other viruses from this family, is a single-stranded, positive-sense ribonucleic acid (RNA) virus. It means that its RNA is equivalent to mammalian mRNA and can be directly translated, in this case, into a single polyprotein, the processing of which generates 3 structural and 7 nonstructural proteins, including the highly conserved, nonstructural glycoprotein NS1, which plays an important role in the replication of its genome and is typical for all flaviviruses. Additionally, this protein may be responsible for at least some neurotoxic effects as it is able to cause reactive oxygen species (ROS) generation, although the expression of NF-E2-related factor 2 (Nrf2) and antioxidant response element (ARE)-dependent genes is also increased by it. On the other hand, at a later stage of the infection, NS1 is secreted and bound with proteins of the complement system, which inactivates them and leads to an impairment of the immune reaction [3,4].

As TBEV is usually transmitted during a tick bite, infection begins in the skin. After introduction of the virus by an infected tick, the virus replicates locally in the Langerhans cells, which are dendritic cells found in the skin. These cells migrate to the lymph nodes, where the virus replicates again. Then, through the lymphatic and blood vessels, the virus is transported to the reticuloendothelial system (spleen, liver, and bone marrow). The virus is also able to cross the blood-brain barrier, but the exact mechanism behind this process is still unknown. Most of the infected patients are able to eliminate the virus through immune mechanisms based mainly on the activity of cytotoxic lymphocytes. Still, the immune system reacts to the virus and an innate immune response begins when the virus is detected by pattern recognition receptors, especially Toll-like receptors in the case of RNA viruses. The first step is the production of interferons, which not only prevents virus’s replication but also activates further defence mechanisms within the body. As tick saliva inhibits NK functions, these cells are believed to be less important, especially in earlier stages of disease [5,6]. Therefore, dendritic cells, which are antigen-presenting cells that present TBEV antigen by MHC II, together with co-stimulatory molecules CD40, CD80, and CD86, leading to the activation of cells from the adaptive immune system, seem to be more important. Activation of the adaptive immune system is also increased by the production of cytokines (IL-1β, TNFα, IL-6, and IL-8) by dendritic cells as well as astrocytes [7,8]. In the adaptive response to TBEV, humoral and cellular responses are observed. Activation of the humoral immune system leads to the production of specific antibodies by B lymphocytes, whereas cellular response leads to the activation of Th and Tc lymphocytes. Antibodies are effective antiviral factors as well as reliable diagnostic markers, but it takes a lot of time before their production is sufficient. The lymphocytes are responsible for regulation of the immune system, mainly by the production of further cytokines. The main function of Tc lymphocytes is the killing of infected cells, which prevents virus replication and infection of other cells. Nevertheless, in the case of neurons, which cannot replicate, this is a substantial problem as these reactions can contribute to serious damage to the CNS. However, in many cases, the immune system is able to eliminate the virus before it penetrates the brain, so the patient does not develop neuroinfection and subsequent CNS damage [5].

It is known that in response to the invasion of pathogens, the host organism activates leukocytes, which produce large amounts of reactive oxygen and nitrogen species (ROS/ RNS), which enhance the host’s immune defence against pathogens [9]. Moreover, infection is usually accompanied by a reduction in the effectiveness of endogenous antioxidant defence mechanisms, and this situation favours the development of oxidative stress [10,11,12]. Disruption of redox homeostasis by pathogens leads to modification of both cellular metabolism and intra- and extracellular signalling. Evidence of oxidative stress in the human body has been found in many viral diseases, including the following: viral hepatitis, AIDS, and human papillomavirus infection [13,14]. Oxidative stress plays a double role in infectious diseases, because ROS modifies both the susceptibility and resistance of the host organism to infections [14,15]. However, overproduction of ROS promotes cytotoxicity and damage to the host molecules/cells/organs, including biological fluids [14,16], but it has also been suggested that ROS participates in inter-pathogenic competition, which protects the body against multi-pathogenic infection [17]. Ticks, regardless of transmitted viruses, can also carry various pathogenic bacteria such as *B*. *burgdorferi*, *A. phagocytophilum*, *C. burnetii*, and *R. conorii*, which may lead to co-infection in the host’s organism. It can be difficult to diagnose and distinguish among others due to the non-specific symptoms and the fact that their diagnosis is based on serology and/or molecular tests, instead of microbiological culture, as is typical in the case of other bacteria. However, the relatively rare detection of such co-infections limits the level of information on the pathomechanisms accompanying the development of the viral disease in such specific cases. 

Therefore, in order to broaden the knowledge of the pathophysiological consequences of TBEV infection and bacterial co-infections (*B. burgdorferi* and *A. phagocytophilum*), the aim of this study is to evaluate the severity of redox imbalance and its consequences for phospholipids and proteins necessary for the physiological functioning of the body. The results of this study also take into account the metabolic response of the organism after treatment and the resolution of clinical symptoms.

## 2. Results

The results of this study show that TBEV infection as well as viral-bacterial co-infection (TBEV+LD, TBEV+HGA—marked, in figures and in their descriptions, for the sake of simplicity only as TBE+LD) increased the activity of one of the basic pro-oxidative enzymes—xanthine oxidase. The consequence was the observed increase in ROS levels both in whole blood and in the granulocytic and lymphocytic fractions (Figure 1). Viral-bacterial co-infection does not significantly modify the pro-oxidative response of the patient’s body compared to TBEV infection. On the other hand, both xanthine oxidase activity and ROS levels were reduced during recovery, which was especially visible in granulocytes. 

Due to the increased ROS generation as a result of infection with pathogens and the intensification of pro-oxidative conditions, the effectiveness of the factor Nrf2, responsible for the transcription of cytoprotective proteins, including antioxidants, is important for the host organism. The results of this study indicated that the expression of the Nrf2 transcription factor as well as its active phosphorylated form (pNrf2) was significantly increased in response to the infections (Figure 2). Despite the increase in the expression of the cytosolic Nrf2 inhibitor—KEAP1 protein, the effectiveness of Nrf2 assessed by the level of the recognised effectiveness index of Nrf2-heme oxygenase-1 (HO-1) was significantly increased, which indicated the possibility of increased transcription of other antioxidant proteins as well.

Despite the increased effectiveness of the transcriptional activity of Nrf2 as a result of infection, not all examined antioxidant enzymes were characterised by increased activity. Both TBEV infection as well as TBEV-bacterial co-infections reduced the activity of enzymes related to the glutathione system and responsible for the protection of, among others, phospholipids such as glutathione peroxidase (GPx) and reductase (GSSGR). It was accompanied by a decreased level of GSH and an increased level of glutathione disulfide (Figure 3). The decrease in the activity of the above-mentioned enzymes observed in this study could have been the result of oxidative modifications to the structures of the newly synthesised proteins. Consequently, despite the elevated levels of enzyme proteins, the observed activity was diminished. On the other hand, therapy (neither symptomatic nor antibiotic) did not significantly change the above-mentioned parameters, except for a reduction in the level of oxidized glutathione.

In contrast to the observed decrease in the activity of the antioxidant enzymes of the glutathione system during TBEV infection and co-infections with bacteria, there was an increase in the activity of superoxide dismutase, the basic antioxidant enzyme in both the cellular and extracellular response to pro-oxidative conditions, which is responsible for the dismutation of the first generated superoxide anion radical. A similar direction of changes is applied to thioredoxin reductase, which is involved, together with thioredoxin and the glutathione system, in the antioxidant protection of phospholipids (Figure 4). However, the accompanying reduction in thioredoxin levels still did not allow the system to function effectively. A particularly difficult situation was observed with regard to co-infection when both the Trx level and the TrxR activity decreased. On the other hand, there was no significant change in the activity of enzymes after symptomatic or antibiotic therapy, but in the case of TBEV infection, Trx level increased, and in co-infection, it decreased after therapy.

Regardless of the modification of antioxidant enzyme systems, TBEV infection and co-infections of TBEV with bacteria also caused changes in the non-enzymatic antioxidant system, which include, among others, vitamins A, E, and C (Figure 5). The results of this study indicate that the levels of these antioxidants decreased both during infection and co-infection, while their levels increased after recovery only in patients infected with TBEV. In the case of co-infections, antibiotic therapy did not have a significant effect on the level of vitamins A and C, but was conducive to raising the level of vitamin E.

As a result of both TBEV infection and co-infection, redox balance is shifted towards pro-oxidative conditions in the bodies of TBE patients, which is accompanied by increased oxidative modifications of macromolecules such as lipids and proteins. Consequently, in the blood plasma of patients with both TBEV infection and bacterial co-infection, a reduction in the level of lipid polyunsaturated fatty acids such as α linolenic acid (LA18:3), arachidonic acid (AA20:4), and eicosahexaenoic acid (EPA20:5) was observed (Figure 6); however, no changes in the level of docosahexaenoic acid (DHA22:5) were observed. In patients with TBEV, the levels of the above-mentioned acids increased after recovery, while in co-infection only the EPA level increased despite antibiotic therapy. In other cases, therapy did not increase the level of polyunsaturated fatty acids.

Taking into account the redox balance shift towards oxidative conditions observed in patients, it can be assumed that changes in the level of lipid fatty acids were the result of oxidative modifications of polyunsaturated fatty acids. This is evidenced by the multiple increase in the level of lipid peroxidation products resulting from both oxidative fragmentation (4-HNE) and oxidative cyclization (8-isoPGF2α) (Figure 7). The level of 4-HNE increased approximately three-fold in the plasma of both groups of patients with TBE and co-infections. Level of this low molecular aldehyde in both cases after recovery. On the other hand, the level of 8-isoPGF2α is about 7 times higher in TBE patients and about 4 times higher in patients with co-infections than in healthy subjects. Therapy reduced the level of cyclization product to approx. 70% of the pre-treatment value in patients with TBE and only to about 88% of the pre-treatment value in patients with co-infections. Nevertheless, the level of lipid peroxidation products, even after clinical recovery, remains significantly higher than in healthy subjects.

The infection also resulted in changes in the structure of plasma proteins. A consequence of the increased level of reactive, electrophilic aldehydes, which are products of lipid peroxidation, was an increase in the level of adducts of these aldehydes, and specifically 4-HNE, with proteins in both groups of patients, while therapy reducing the 4-HNE level in the plasma of patients favoured a significant decrease in the level of these adducts, but only in the TBE group (Figure 8).

As a result of the oxidative stress observed in TBE, there were also oxidative modifications of proteins’ structures caused by the direct action of ROS and estimated as an increased level of dityrosine in the case of infection and co-infection and carbonyl groups (CBO) indicated only in the case of TBE (Figure 8). In addition, a reduction in the level of tryptophan was observed as a result of oxidative degradation in both groups of infected patients (TBE and TBE+LD). The level of tryptophan did not significantly change after treatment, but there was a statistically significant reduction in the level of dityrosine in the group of TBE patients after clinical recovery.

A consequence of the oxidative-antioxidant imbalance in the bodies of patients with TBEV infection and bacterial co-infections was the occurrence of oxidative stress in the patients’ blood, which favoured the intensification of the pro-inflammatory response assessed on the basis of the level of the transcription factor NFkB and the product of its transcription activity, cytokine TNFα. It was shown that the expression of NFkB, both subunits, as well as TNFα was significantly increased in the initial phase of infection and persists (or even increases) despite recovery (Figure 9).

The statistical analysis performed in the TBE group before treatment showed a negative correlation between SOD and pleocytosis in CSF (R = −0.35), a negative correlation of NPs with WBC in blood (R = −0.38) and a positive correlation of HNE-b with CRP, pleocytosis, and protein concentration in CSF (R = 0.38; 0.44, 0.32, respectively). After treatment, only a negative correlation of Vitamin C with protein concentration in CSF (R = −0.35) was observed. 

## 3. Discussion 

During viral and bacterial infections, the host organism reacts to signalling molecules that arise in response to the presence of the pathogen [18]. The host organism activates leukocytes, which, as a result of increased activity of pro-oxidative enzymes responsible for the production of ROS, produce large amounts of them and play a key role in the host’s immune defence against pathogens [9]. So far, it has been shown that TBEV infection causes metabolic changes in the host organism, mainly related to the inflammatory reaction and immune modification [19]. The results of this study show that in the group of patients with TBE, there were statistically significant differences in the WBC count, percentage of neutrophils, and CRP concentration before and after treatment. Although CRP is used as an inflammation marker in many conditions, its utility in TBE is limited. This is confirmed by the results of this study showing that in most patients with TBE, CRP in the acute phase of the disease is in the normal range and therefore cannot be used to monitor the course of the disease. Interestingly, in patients with TBE, CRP, WBC, and the percentage of neutrophils in peripheral blood were higher than in patients with TBE co-infection. Therefore, basic laboratory tests are of limited use in the differential diagnosis of tick-borne diseases, and there is a need to analyse other biochemical parameters.

One of the basic conditions for the physiological state of the human body is to maintain redox balance both at the level of cells and tissues, including body fluids [20,21]. However, in response to the invasion of the pathogen, the host organism activates leukocytes which produce large amounts of ROS/RNS involved in the host’s immune defence [16]. It is known that the increased generation of ROS in biological fluids as a result of infection is mainly the result of reactions catalysed by xanthine oxidase, myeloperoxidase, and nitric oxide synthase [22,23], which modulate the signalling cascade [24]. In this study, in the blood plasma of patients with both TBEV infection and bacterial co-infection, a significant increase in the activity of plasma xanthine oxidase and an accompanying increase in the level of ROS both in whole blood and in granulocytes and lymphocytes were found. However, since it is suggested that ROS are involved in the degradation of pathogens, they may limit their effect on the human body. On the other hand, they may also take part in inter-pathogenic competition [17]. This mechanism may explain the modest differences in ROS levels observed in this study, as well as the relatively modest metabolic changes resulting from bacterial TBEV co-infection compared to TBEV infection alone, including less stimulation of granulocytes by pro-inflammatory cytokines, leading to their lower activation. On the other hand, antibiotics used to treat patients by killing bacteria reduce the level of LPS supplied by them, which are powerful activators of neutrophils, and consequently reduce the activation of neutrophils and, consequently the ROS generation, observed in this study.

Regardless of the beneficial role of ROS in the elimination of pathogens, their overproduction favours the reduction of antioxidant capacity in the plasma of patients with TBE and bacterial co-infections and, consequently, the formation of oxidative stress, as shown by the results of this study. This is due to the reduced efficiency of the components of the glutathione system by reducing the activity of glutathione peroxidase and reductase and the level of GSH, as well as the effectiveness of the thioredoxin system as a result of reducing the level of Trx. It is known that both systems work together to protect cells against the effects of ROS overproduction, especially against lipid oxidative modifications [25,26], which is not effective in both TBEV infection as well as bacterial co-infections. Moreover, the reduction of GSH levels promotes viral replication [27], which may aggravate the disease process. Changes in the level/activity of the above antioxidants are also accompanied by a decrease in the levels of the other non-enzymatic antioxidants, including vitamins A, E, and C, that work together with glutathione in the protection of lipids [28] (Figure 10). However, their decreased levels observed in TBE infection and co-infections confirm the reduction of their protective effects. In such a situation, even the increased activity of superoxide dismutase and thioredoxin reductase observed in the plasma of patients with the above-mentioned infections is not able to ensure the effectiveness of antioxidant actions.

The reduced effectiveness of the antioxidant protection of polyunsaturated fatty acids (PUFAs) in phospholipids with the increase in the ROS level promotes lipid peroxidation assessed in this study by lowering the level of PUFAs and increasing the level of lipid peroxidation products in the plasma of patients with TBE as well as TBEV and bacterial infections. Levels of PUFAs such as LA(18:3), AA(20:4), and EPA(20:5) assessed in the plasma of patients in the course of TBEV infection and bacterial co-infections are significantly reduced, but their levels return to control values after treatment, which may indicate the effectiveness of the treatment. It is known that the lipid metabolism is controlled both directly and indirectly by ROS and/or lipid metabolising enzymes, mainly phosphatases, cyclooxygenases, and lipoxygenases, the activity of which usually increases under the influence of oxidative stress [29].

ROS-dependent lipid peroxidation is based on free radical chain reactions with PUFAs, initiated mainly by hydroxyl or hydroperoxide radicals generated by pro-oxidative enzymes [29], whose activity is enhanced in TBE and co-infections. Products of peroxidation are formed in the oxidative fragmentation or cyclization of PUFAs hydrocarbon chains. As a result of cyclization, prostaglandin derivatives with a five-membered prostane ring, such as isoprostanes (mainly assessed as 8-iso-PGF2α) are generated, which are then released from lipid structures by phospholipase A2, the activity of which is usually exacerbated by pro-oxidative conditions observed in the plasma of patients with TBE [29,30]. It was also shown that 8-iso-PGF2α is involved in the pathogenesis of different infectious diseases [31,32]. Since isoprostanes also play a role in pro-inflammatory processes, their elevated levels in the plasma of patients with TBE indicate an increase in systemic inflammation aimed at fighting infection. On the other hand, in the case of bacterial co-infection, a lower level of 8-iso-PGF2α than in TBE is observed, which indicates an impairment of this signalling pathway, and thus a lower ability of the body to fight pathogens. However, in the process of lipid peroxidation, there are also oxidative fragmentation reactions of PUFAs hydrocarbon chains with the formation of α,β−unsaturated reactive aldehydes, including 4-HNE, the level of which is significantly elevated in TBE and bacterial co-infections. Electrophilic 4-HNE forms adducts with most of the nucleophilic components of the organism, including amino acid residues of proteins such as cysteine, histidine, and lysine [33], disrupting the metabolic processes in which modified proteins participate [34,35]. 4-HNE forms adducts with antioxidant enzymes that may reduce its antioxidant effectiveness [36,37]. It is very likely in the disease cases analysed in this study, as the level of 4-HNE-protein adducts is significantly elevated. This mainly concerns glutathione peroxidase, the activity of which is clearly reduced in the plasma of patients with TBE infections and bacterial co-infections. However, 4-HNE can also stimulate an antioxidant response, positively influencing the activity of the transcription factor Nrf2. Under physiological conditions, Nrf2, which is responsible for the biosynthesis of cytoprotective proteins, including antioxidants, is in a complex with its cytosolic inhibitor protein, Keap1, which directs the transcription factor to ubiquitination and proteasomal degradation [38]. However, under oxidative conditions, both ROS and lipid peroxidation products, reacting with the critical Keap1 cysteine residue, inactivate this inhibitor [39], which leads to the activation of Nrf2, which regulates the transcription of genes, including those encoding proteins involved in the maintenance of redox balance, detoxification, repair of macromolecular damage, and metabolic balance [40]. The increased antioxidant gene transcription is evidenced by the significantly increased level of heme oxygenase, which is considered to be the primary indicator of this transcription factor’s effectiveness [38]. On the other hand, the decrease in 4-HNE observed after treatment may be due to its interaction with the N-terminal amino acid of c-Jun kinase (JNK), leading to its activation and translocation into the nucleus [41,42], which may also promote activation of Nrf2 in antioxidant response. Earlier, it was found that the intensification of transcriptional activity of antioxidant genes by Nrf2 correlates with resistance to HSV1 infection [43], so a similar situation may apply to tick-borne encephalitis. In inflammation, including the observed TBEV infection and bacterial co-infections, Nrf2 activation ultimately restores redox homeostasis by upregulating essential antioxidants, including heme oxygenase 1 (HO-1), which is known to be a component of effective TBE cell resistance to numerous pathogens [44], but also superoxide dismutase and thioredoxin reductase. This explains why, in response to the introduction of the pathogen into the human body and the resulting increased production of ROS, it is usually accompanied by an overproduction of antioxidants [45,46]. It is known, however, that the molecular mechanisms of resistance modeled by Nrf2 in response to stress may vary with the type of pathogen as well as with the immune response of the host.

Regardless of the direct action of elements of the antioxidant system, such as Trx or GSH, the level of which is lowered as an observed result of TBEV infection, their metabolic importance in the body is also associated with the role of modifiers of the activity of the transcription factor NF-κB, which is responsible for the pro-inflammatory response of the host organism [47]. It is known that Trx is responsible for the reduction of Cys62 NFkB, necessary for the biosynthesis of proinflammatory proteins [48]; therefore, the reduced level of Trx in the plasma of patients with TBEV infections and bacterial co-infections may indicate a partially reduced transcriptional activity of NFkB. Such a mechanism of Trx action in TBEV infection may explain the observed illogical tendency to increase the level of NFkB subunits and the level of the product of its transcriptional activity, the pro-inflammatory cytokine (TNFα), after recovery, accompanied by a tendency to increase in Trx level. On the other hand, in the case of co-infections, there is a tendency to further decrease the Trx level after antibiotic therapy and, consequently, a decrease in the TNFα level. However, the transcriptional and signalling activity of NFkB also depends on the level of GSH, which modifies the structure of this transcription factor through glutathionylation of cysteine 62, and thus inhibits its pro-inflammatory effects [49]. Proteomic studies showed increased formation of GSH protein adducts in the plasma of patients with TBE, as well as with co-infections [unpublished data]. This may be the reason for the decreased plasma GSH levels in TBE patients. Similar results are seen with some bacterial infections, including those caused by *B. burgdorferi*, which induce protein glutathionylation in the host organism [50]. GSH may glutationylate IKK—an NFkB inhibitor in the plasma of patients with TBE, especially in the case of co-infections, which indicates the possibility of increasing the effectiveness of this transcription factor. Glutathionylation by inducing the activation of the NFκB pathway causes an increase in systemic inflammation, much stronger in co-infected patients, when GSH modifies IKK more strongly than in TBE patients. Additionally, high levels of kinases, including PKA, PKC, and CDK4, are observed in the plasma of patients infected with tick-borne diseases. This disrupts the phosphorylation-dependent signalling pathways. Protein glutathionylation is a major redox immune mechanism that affects the function of not only NF-kB but also proteins such as STAT3, PKA, TRAF3, and TRAF6 [18]. Therefore, by inhibiting NF-kB-mediated signalling, GSH may also play an anti-inflammatory role and exert a protective effect in the infections [51]. However, the results described in this study additionally complement this knowledge with the information that the level of protein carbonyl groups in the case of TBEV co-infection is significantly lower compared to the level of CBO in the plasma of TBE patients, hence the induction of NFkB activity may be lower regardless of the level of its subunits similar in both groups of patients. On the other hand, 4-HNE, the level of which is elevated in the plasma of TBE and co-infected patients, affects both direct and indirect pro-inflammatory signalling. By binding to the cytoplasmic NFκB inhibitor—IkB, 4-HNE may increase the transcriptional activity of NFκB, which is manifested by an increased level of TNFα [52], which by binding with appropriate receptors, stimulates the expression of NFκB. It is also known that the interaction of 4-HNE with TLR prevents the activation of this transmembrane protein, which suppresses the immune functions that initiate NFκB activation during bacterial infection [53]. 

## 4. Materials and Methods

### 4.1. Samples Collection

Blood samples were obtained from 40 patients with tick-borne encephalitis (14 female and 26 male), mean age: 40 (23–58) and 6 patients with TBE co-infection with other tick-borne pathogens including *Borrelia burgdorferi (Bb)* (Lyme disease—LD) and *Anaplasma phagocytophilum (Ap)* (human granulocytic anaplasmosis—HGA) (4 female and 2 male), mean age: 42 (22–63) treated in the Department of Infectious Diseases and Neuroinfections, Medical University of Bialystok, Poland. 

TBE was diagnosed according to European Academy of Neurology (EAN) guidelines [54], based on clinical symptoms, positive serology and lymphocytic pleocytosis in the cerebrospinal fluid (CSF). Co-infection was diagnosed if one patient was infected with at least two different pathogens. LD were defined on the basis of clinical presentation of erythema migrans or fulfilled criteria for neuroborreliosis [55,56]. HGA was diagnosed according to the case definition by CDC when all 3 criteria were fulfilled (https://wwwn.cdc.gov/nndss/conditions/ehrlichiosis-and-anaplasmosis/case-definition, accessed on 11 November 2019).

Co-infection was diagnosed if one patient was infected with at least two different pathogens. LD was defined on the basis of clinical presentation of erythema migrans (1 patient) or fulfilling criteria for neuroborreliosis (1 patient). Moreover, in two patients anti-*Borrelia burgdorferi* IgM and IgG antibodies were detected and the patients were treated for probable cases of neuroborreliosis. Infection was associated with *Borrelia burgdorferi* sensu lato. In four cases human granulocytic anaplasmosis based on clinical picture, laboratory tests and PCR was diagnosed. 

The control group comprised of 20 healthy donors (7 female and 13 male, mean age: 41 (28–55 years old)). The study was conducted in accordance with the Declaration of Helsinki, and the protocol for the collection of all blood samples was approved by the Local Bioethics Committee in Medical University of Bialystok (Poland), No. R-I-002/169/2018. Written informed consent was obtained from all participants.

The blood was taken twice as follows: at admission and after treatment. The mean time between these two examinations in TBE patients was 29.4 ± 10.99 days, min. 7 days, max 52 days, while the mean time between these two examinations in co-infected patients was 32.8 ± 4.21 days, min. 28 days, max 39 days.

Blood samples were collected into ethylenediaminetetraacetic acid (EDTA) tubes and a two-stage centrifugation was carried out. In the first stage, the sample was centrifuged at 3000× *g* (4 °C) to separate the plasma, buffy coat, and erythrocytes. To obtain clear fraction of granulocytes, obtained buffy coat was layered on Gradisol G (Aqua-Med ZPAM–KOLASA, Łódź, Poland) and subjected to 25 min centrifugation at 300× *g* at room temperature. To obtain clear fraction of lymphocytes whole blood was layered on Gradisol L (Aqua-Med ZPAM–KOLASA, Łódź, Poland) and centrifuged for 25 min at 300× *g*. The lymphocyte containing fraction was collected from above the Gradisol layer, washed three times in PBS (3 min centrifugation at 300× *g*) and resuspended in PBS containing a protease inhibitor mix. Cells in the middle fraction were collected, washed, and re-suspended in PBS containing a proteasome inhibitor mix. The purity of the obtained cell fraction was examined microscopically (Nikon Eclipse Ti, Nikon Instruments Inc., New York, NY, USA). Butylhydroxytoluene (BHT), as an antioxidant was added to all samples (plasma, granulocytes, and lymphocytes) to prevent oxidation. Samples were stored at –80 °C until analysis.

Demographic and clinical characteristics of patients and control, as well as comparison of their laboratory data are presented in Table 1 and Table 2.

### 4.2. Methods

#### 4.2.1. Pro-Oxidant Parameters

##### Pro-Oxidant Enzyme Activity

Xanthine oxidase (XO; EC 1.17.3.2) activity was estimated in plasma as the rate of uric acid generation from xanthine detecting at a wavelength of 292 nm [57]. One unit of xanthine oxidase activity was defined as 1 µmol uric acid produced per minute (37 °C). Enzyme activity was expressed as units per ml of plasma.

##### ROS Level Determination

50 μL of blood and granulocytes/lymphocytes obtained from 2 mL of blood collected in heparinized capillary tubes were analyzed. Among spin trapping (otherwise labelled probe) molecules, suitable for biological utilization, 1-hydroxy-3-methoxycarbonyl-2,2,5,5-tetramethylpyrrolidine (CMH, Noxygen Science Transfer & Diagnostics, Germany) was adopted. A 400 µM CMH solution was prepared in buffer (Krebs-Hepes buffer (KHB) containing 25 μm deferoxamine methane-sulfonate salt (DF) chelating agent and 5 μm sodium diethyldithiocarbamate trihydrate (DETC) at pH 7.4. Blood and granulocytes/lymphocytes in PBS were immediately treated with CMH (1:1) [58]. 50 μL of the obtained solution was put in the glass EPR capillary tube that was placed inside the cavity of the e-scan spectrometer (Noxygen GmbH/Bruker Biospin GmbH, Rheinstetten, Germany) for data acquisition. The oxidation of CMH leads to the formation of the paramagnetic 3-methoxycarbonyl-proxyl nitroxide (CM•). ROS levels are expressed in nmol/mL.

#### 4.2.2. Antioxidant Parameters

##### Determination of Protein Antioxidants

The activity of glutathione peroxidase (GSHPx–EC.1.11.1.6) in plasma was measured spectrophotometrically (340 nm) using the method of Paglia and Valentine [59] through estimating the conversion of NADPH to NADP+. One unit of GSH-Px activity was defined as the amount of enzyme catalyzing the oxidation of 1 µmol NADPH min^−1^ at pH 7.4, 25 °C. Enzyme activity was expressed as units per ml of plasma.

The activity of glutathione reductase (GSSGR–EC.1.6.4.2) in plasma was measured spectrophotometrically (340 nm) according to the method of Mize and Longdon [60] through monitoring the oxidation of NADPH to NADP+. One unit of GSSGR oxidised 1 µmol of NADPH/min at pH 7.4, 25 °C. Enzyme activity was expressed as units per ml of plasma.

The activity of superoxide dismutase (Cu,Zn–SOD– EC.1.15.1.1) activity in plasma was determined spectrophotometrically (480 nm) according to the method of Misra and Fridovich [61] modified by Sykes [62]. One unit of Cu,Zn–SOD was defined as the amount of the enzyme which inhibits epinephrine oxidation to adrenochrome by 50%. Enzyme specific activity was expressed as units per ml of plasma.

Thioredoxin reductase (TrxR-EC.1.8.1.9) activity in plasma was estimated using commercial assay kit (Sigma-Aldrich, St. Louis, MO, USA). The assay was based on reduction of 5,5′-dithiobis(2-nitrobenzoic) acid by NADPH to 5-thio-2-nitrobenzoic acid which was estimated by colorimetric measurement at 412 nm [63]. Obtained data were expressed as units per ml of plasma.

Thioredoxin (Trx) level in plasma was quantified using ELISA method [64]. ELISA plates (Nunc Immuno Maxisorp, Thermo Scientific, Waltham, MA, USA) with samples were incubated overnight with primary antibody against thioredoxin (Abcam, Cambridge, MA, USA) and for 1 h with secondary goat anti-rabbit/mouse EnVision+ Dual Link/HRP solution (1:100) (Agilent Technologies, Santa Clara, CA, USA). Then 0.1 mg/mL TMB in citric buffer with 0.012% H_2_O_2_ as a chromogen was added. The reaction was stopped by 2 M sulfuric acid and absorption was read at 450 nm with 620 nm as a reference filter. The Trx level was expressed as micrograms per ml of plasma.

##### Determination of Low Molecular Antioxidants

Reduced GSH content in plasma was measured according to the procedure of Maeso using capillary electrophoresis [65]. The separation was performed on a fused-silica capillary with a 40 cm effective length at a constant voltage of 27 kV. 2.2 [75 µm (i.d.) × 40 cm (total length)/10 cm (length to detector)] with a spectrophotometer detection at 200 nm. The reduced glutathione level was expressed as nanomoles per ml of plasma.

The HPLC methods were used to determine the levels of vitamins A and E [66] and vitamin C [67]. The extraction of vitamin A and E from plasma was conducted with hexane containing 0.025% butylated hydroxytoluene. The hexane phase was removed and dried, and 50 μL of the hexane extract was injected into the column. For ascorbic acid determination, plasma and an equal volume of metaphosphoric acid were mixed. Samples were centrifuged (1000× *g*, 10 min) before the analysis to remove residues of precipitated proteins. After centrifuging the samples were immediately analyzed. Vitamin content was analysed on HPLC system (Agilent Technologies, Santa Clara, CA, USA) with a diode array detector (294 nm) using a RP C18 column for vitamins A and E, while a RP C18 column and UV detection at 250 nm for vitamin C determination was applied. The vitamin concentration was expressed in nanomoles per ml of plasma.

#### 4.2.3. Phospholipid Metabolism

##### Phospholipid Fatty Acids Profile

The phospholipid fatty acids as fatty acid methyl esters (FAMEs) were determined by gas chromatography [68]. Lipid fraction was isolated by Folch extraction using chloroform/methanol mixture (2:1, *v*/*v*) with 0.01% butylated hydroxytoluene. For phospholipid fatty acid separation the thin layer chromatography technique was used with heptane–diisopropyl ether–acetic acid (60:40:3, *v*/*v*/*v*) as a mobile phase. Next the phospholipid fatty acids were transmethylated to fatty acid methyl esters using boron trifluoride in methanol. FAMEs were analysed by gas chromatography with a flame ionisation detector (FID) using Clarus 500 Gas Chromatograph (Perkin Elmer, Waltham, MA, USA). The separation of sample components was performed by capillary column with Varian CP-Sil 88 stationary phase (50 m × 0.25 mm, ID 0.2 μm, Varian). The qualitative analysis was based on a comparison of retention time of obtained FAMEs with authentic standards and the quantitative analysis was performed using the internal standard method with nonadecanoic acid (19:0) and 1.2-dinonadecanoyl-sn-glycero-3-phosphocholine (19:0 PC) as internal standards.

##### Lipid Peroxidation Products

Product of phospholipid fragmentation (low molecular aldehyde—4-HNE) were measured by GC/MS in plasma as the *O*-PFB-oxime-TMS derivative, using minor modifications of the method of Tsikas et al. [69]. d_3_-4-HNE was added as an internal standard, and aldehydes were derivatized by the addition of *O*-(2,3,4,5,6-pentafluoro-benzyl) hydroxylamine hydrochloride (0.05 M in PIPES buffer, 200 μL) and incubated for 24 h at room temperature. After incubation, samples were deproteinized by the addition of 1 mL of methanol, and *O*-PFB-oxime aldehyde derivative was extracted by the addition of 2 mL of hexane. The top hexane layer was transferred into borosilicate tubes, and evaporated under a stream of argon gas, followed by the addition of *N,O*-bis(trimethylsilyl)trifluoroacetamide in 1% trimethylchlorosilane. A 1 μL aliquot was injected into the column. The 4-HNE content was determined by 7890A GC–7000 quadrupole MS/MS (Agilent Technologies, Santa Clara, CA, USA) with selected ion monitoring mode. The used ions were as follows: *m*/*z* 2420 for 4-HNE-PFB-TMS and *m*/*z* 2040 for IS derivatives. 4-HNE was expressed as nanomoles per ml of plasma.

The determination of product of phospholipid cyclisation [total F_2_-isoprostanes (8-isoPGF_2α_)] was based on method of Coolen [70]. Solid phase extraction was used for F2-isoprostane isolation. Data were obtained with an Agilent 1290 UPLC system with an Agilent 6460 QqQ spectrometer with electrospray ionisation source (ESI). 8-isoPGF_2α_ was analysed in negative-ion mode using the following MRM mode: *m*/*z* 353.2→193.1 (for 8-isoPGF_2α_) and 357.2→197.1 (for 8-isoPGF_2 α_-d_4_).

#### 4.2.4. Determination of Protein Expression

Protein expression measurement was performed using enzyme-linked immunosorbent assay (ELISA) [71]. Lysates of cells (granulocytes and lymphocytes) were applied to ELISA plate wells (Nunc Immuno MaxiSorp, Thermo Scientific, Waltham, MA, USA). Plates with attached proteins were incubated at 4 °C for 3 h with blocking solution (5% fat-free dry milk in carbonate binding buffer). After washing with PBS supplemented with 0.1% Tween 20, samples were incubated at 4°C overnight with appropriate primary antibody against NFκB (p52 or p62), TNFα, HO-1 (host: mouse) (Sigma-Aldrich, St. Louis, MO, USA), Keap1, Nrf2, phospho-Nrf2 (Ser40) (host: rabbit) (Santa Cruz Biotechnology, CA, USA). All antibodies were used at a concentration of 1:1000. Next, following washing (PBS supplemented with 0.1% Tween 20), plates were incubated for 30 min with peroxidase blocking solution (3% H_2_O_2_, 3% fat free dry milk in PBS) at room temperature. As a secondary antibody goat anti-rabbit/mouse EnVision+ Dual Link/HRP solution (1:100) (Agilent Technologies, Santa Clara, CA, USA) was used. After 1 h of incubation at room temperature, secondary antibodies were removed and plates were incubated with chromogen substrate solution (0.1 mg/mL TMB, 0.012% H_2_O_2_) for 40 min. The reaction was stopped by adding 2 M sulfuric acid and absorption was read within 10 min at 450 nm and automatically recalculated from standard curves for each protein (NFκBp52; Lifespan Biosciences, Seattle, WA, USA, NFκBp65; OriGene Technologies, Rockville, USA, TNFα; Merck, Darmstadt, Germany, Nrf2; MyBioSource, San Diego, CA, USA, pNRF2; human Nrf2 (phospho S40), Abcam, Cambridge, GB, Keap1; Sino Biological, Eschborn, Germany and HO-1; Enzo Life Sciences, Ann Arbor, MI, USA) into the protein level in the samples.

#### 4.2.5. Determination of Protein Oxidative Modifications

Protein oxidative modifications in granulocytes were estimated at tryptophan and protein carbonyl group levels. To analyse tryptophan levels, samples were diluted in 0.1 mol/L H_2_SO_4_ (1:10), and fluorescence emission/excitation at 325 nm/420 nm and 288 nm/338 nm, respectively, was measured [72]. All results were normalised for 1 milligramme of protein. The level of protein carbonyl groups was determined spectrophotometrically (370 nm) using 2,4-dinitrophenylhydrazine [73] and was expressed as nanomoles of carbonyl groups per milligram of protein.

The 4-HNE-protein adducts level in granulocytes was measured by ELISA method [74] using anti-4-HNE-protein monoclonal mouse antibody (Invitrogen, Burlington, Canada) and goat anti-rabbit/mouse EnVision+ Dual Link/HRP solution (1:100) (Agilent Technologies, Santa Clara, California, USA) as secondary antibodies. The concentrations of 4-HNE–protein adducts were determined using a calibration curve range of 0.5–25 pmol/mg of BSE (r^2^—0.9983) and normalised for 1 milligramme of protein.

#### 4.2.6. Statistical Analysis

Data were expressed as mean ± SD and were analysed by one-way analysis of variance (ANOVA) followed by a post hoc Tukey testing using Statistica software (Statistica 13.3, StatSoft, Krakow, Poland). Results were compared using Mann–Whitney U test and Wilcoxon signed-rank test. Values of *p* ≤ 0.05 were considered significant, and only these results were discussed in detail.

Correlation between the levels of oxidative stress markers and other markers of inflammation was calculated by use of Spearman rank correlation test.

## 5. Conclusions

The growing number of TBEV infections and co-infections causes both diagnostic and therapeutic problems. This is evidenced by relatively small differences in metabolic parameters in both groups of patients, both before and after pharmacotherapy. In addition, slight differences in the blood of patients infected only with the virus and the virus and bacteria may indicate the effect of inter-pathogenic competition. However, some metabolic reactions differentiate classic TBEV infection from co-infections. This applies in particular to the increased activity of TrxR in the case of TBE and a decrease in the case of TBEV co-infection, which can be used in differential diagnosis and therapy. In addition, the reduced levels of antioxidants (Trx and GSH) and the increased concentration of lipid peroxidation products (4-HNE and 8-iso-PGF2α) promote the modifications of the structure and function of proteins, e.g., transcription factors (Nrf2 and NFkB) model their effectiveness. On the other hand, increasing the level of Trx after therapy may contribute to the intensification of the inflammatory process. In turn, the lower level of 8-iso-PGF2α observed in co-infections indicates an impairment of the body’s ability to intensify inflammation and respond to infection. Therefore, in order to propose a new perspective, both diagnostic and therapeutic, there is a further need to analyse metabolic changes in larger groups of patients with TBE and co-infections.

## Figures and Tables

**Figure 1 pathogens-11-00384-f001:**
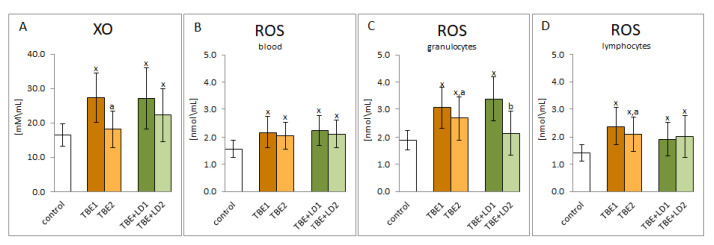
The effect of TBEV infection (TBE1) and TBEV+Bb/Ap co-infection (TBE+LD1) as well as therapy (TBE2 and TBE+LD2) on the pro-oxidative response of the patient’s organism, visible in the activity of plasma xanthine oxidase (XO) (**A**) as well as in the level of ROS assessed in plasma (**B**), granulocytes (**C**) and lymphocytes (**D**) compared to healthy subjects (control). The mean values for TBE patients (before–TBE1 and after treatment–TBE2, *n* = 40), patients with TBE+LD (before–TBE+LD1 and after treatment–TBE+LD2, *n* = 6) and control subjects (*n* = 20) with SD are shown as follows: x—differences vs. control group, *p* < 0.05; a—differences vs. TBE1 group, *p* < 0.05; b—differences vs. TBE+LD1 group, *p* < 0.05.

**Figure 2 pathogens-11-00384-f002:**
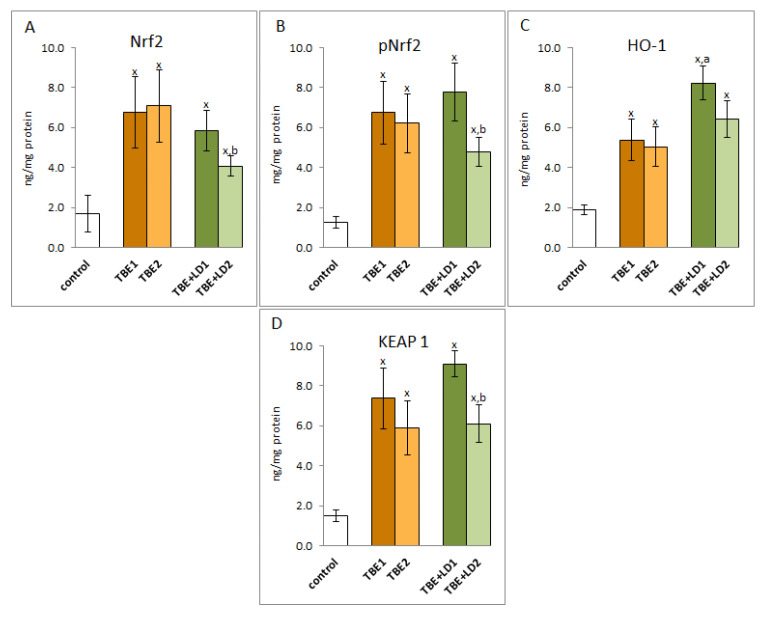
The effect of TBEV infection (TBE1) and TBEV+Bb/Ap co-infection (TBE+LD1) as well as therapy (TBE2 and TBE+LD2) on the effectiveness of the Nrf2 (**A**) transcription factor, visible in the expression of Nrf2/pNrf2 (**B**), HO-1 (**C**) and KEAP1 (**D**) assessed in granulocytes of patients and compared to healthy subjects (control). The mean values for TBE patients (before–TBE1 and after treatment–TBE2, *n* = 40), patients with TBE+LD (before–TBE+LD1 and after treatment–TBE+LD2, *n* = 6) and control subjects (*n* = 20) with SD are shown as follows: x—differences vs. control group, *p* < 0.05; a—differences vs. TBE1 group, *p* < 0.05; b—differences vs. TBE+LD1 group, *p* < 0.05.

**Figure 3 pathogens-11-00384-f003:**
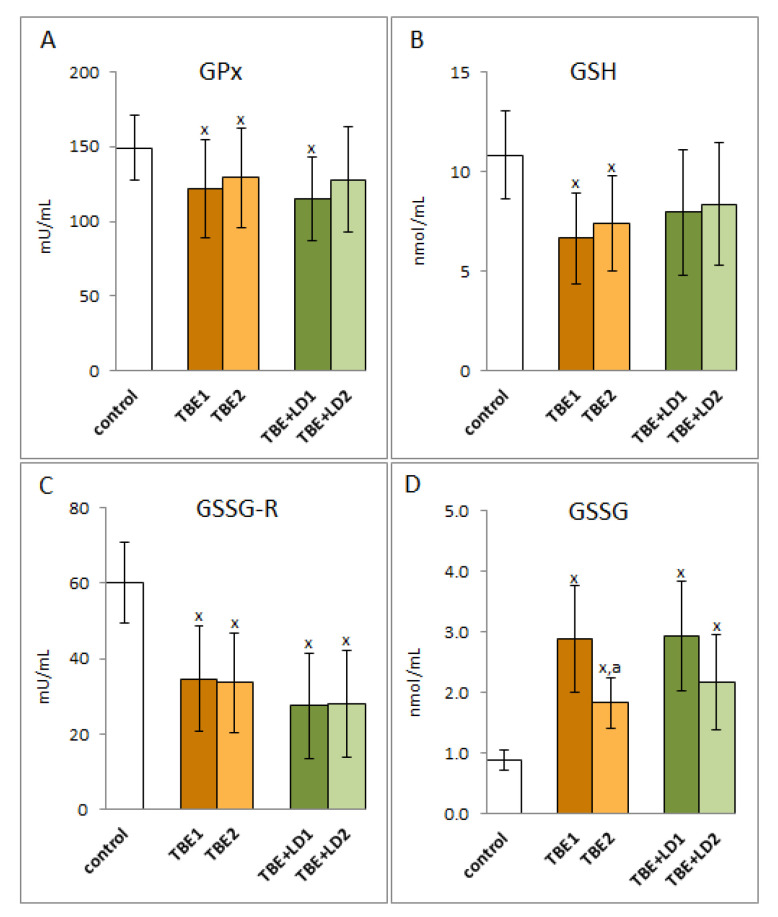
The effect of TBEV infection (TBE1) and TBEV+Bb/Ap co-infection (TBE+LD/HGA1) as well as therapy (TBE2 and TBE+LD2) on the glutathione system of the patient’s organism response, visible in the activity of plasma glutathione peroxidase (GPx) (**A**), level of GSH (**B**) and activity of glutathione reductase (GSSG-R) (**C**) as well as in the level of GSSG (**D**) assessed in plasma compared to healthy subjects (control). The mean values for TBE patients (before–TBE1 and after treatment–TBE2, *n* = 40), patients with TBE+LD (before–TBE+LD1 and after treatment–TBE+LD2, *n* = 6) and control subjects (*n* = 20) with SD are shown as follows: x—differences vs. control group, *p* < 0.05; a—differences vs. TBE1 group, *p* < 0.05.

**Figure 4 pathogens-11-00384-f004:**
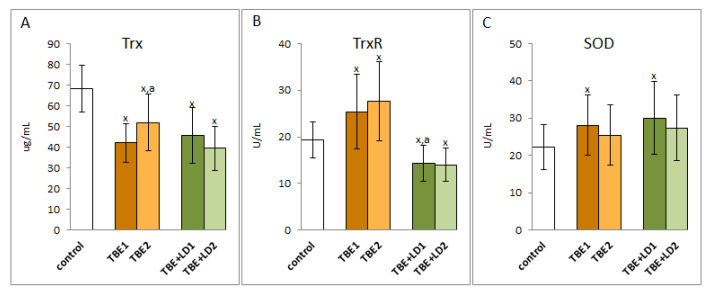
The effect of TBEV infection (TBE1) and TBEV+Bb/Ap co-infection (TBE+LD1) as well as therapy (TBE2 and TBE+LD2) on the thioredoxin system including thioredoxin (Trx) (**A**) and thioredoxin reductase (TrxR) (**B**) as well as superoxide dismutase (SOD) (**C**) in plasma of patients compared to healthy subjects (control). The mean values for TBE patients (before–TBE1 and after treatment–TBE2, *n* = 40), patients with TBE+LD (before–TBE+LD1 and after treatment–TBE+LD2, *n* = 6) and control subjects (*n* = 20) with SD are shown as follows: x—differences vs. control group, *p* < 0.05; a—differences vs. TBE1 group, *p* < 0.05.

**Figure 5 pathogens-11-00384-f005:**
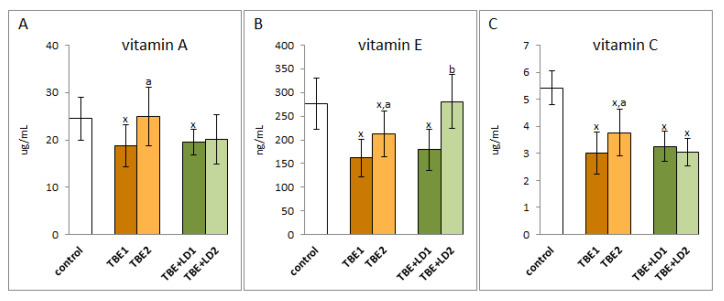
The effect of TBEV infection (TBE1) and TBEV+Bb/Ap co-infection (TBE+LD1) as well as therapy (TBE2 and TBE+LD2) on the vitamins A (**A**), E (**B**) and C (**C**) in plasma of patients compared to healthy subjects (control). The mean values for TBE patients (before–TBE1 and after treatment–TBE2, *n* = 40), patients with TBE+LD (before–TBE+LD1 and after treatment–TBE+LD2, *n* = 6) and control subjects (*n* = 20) with SD are shown as follows: x—differences vs. control group, *p* < 0.05; a—differences vs. TBE1 group, *p* < 0.05; b—differences vs. TBE+LD1 group, *p* < 0.05.

**Figure 6 pathogens-11-00384-f006:**
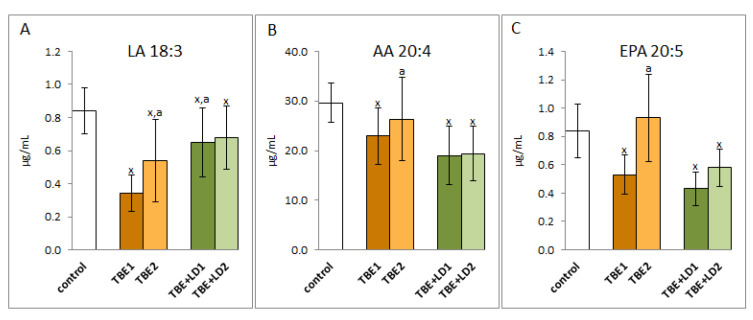
The effect of TBEV infection (TBE1) and TBEV+Bb/Ap co-infection (TBE+LD1) as well as therapy (TBE2 and TBE+LD2) on the lipid polyunsaturated fatty acids (PUFAs), visible in the level of linoleic acid (LA) (**A**), arachidonic acid (AA) (**B**), and eicosapentaenoic acid (EPA) (**C**) assessed in plasma of patients compared to healthy subjects (control). The mean values for TBE patients (before–TBE1 and after treatment–TBE2, *n* = 40), patients with TBE+LD (before–TBE+LD1 and after treatment–TBE+LD2, *n* = 6) and control subjects (*n* = 20) with SD are shown as follows: x—differences vs. control group, *p* < 0.05; a—differences vs. TBE1 group, *p* < 0.05.

**Figure 7 pathogens-11-00384-f007:**
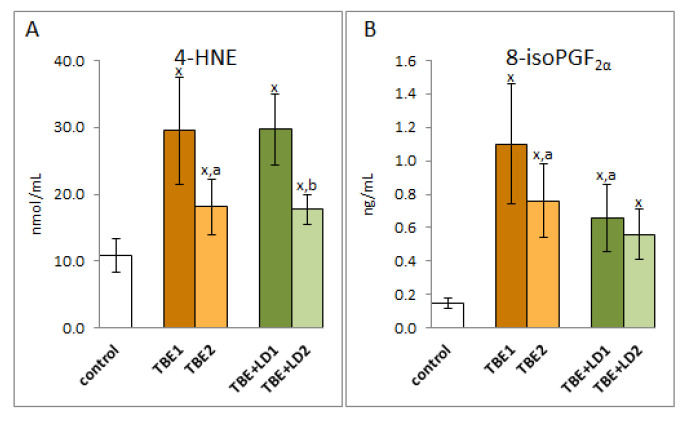
The effect of TBEV infection (TBE1) and TBEV+Bb/Ap co-infection (TBE+LD1) as well as therapy (TBE2 and TBE+LD2) on the lipid peroxidation products visible in the level of 4-HNE (**A**) and 8-isoPGF2α (**B**) assessed in plasma of patients compared to healthy subjects (control). The mean values for TBE patients (before–TBE1 and after treatment–TBE2, *n* = 40), patients with TBE+LD (before–TBE+LD1 and after treatment–TBE+LD2, *n* = 6) and control subjects (*n* = 20) with SD are shown as follows: x—differences vs. control group, *p* < 0.05; a—differences vs. TBE1 group, *p* < 0.05; b—differences vs. TBE+LD1 group, *p* < 0.05.

**Figure 8 pathogens-11-00384-f008:**
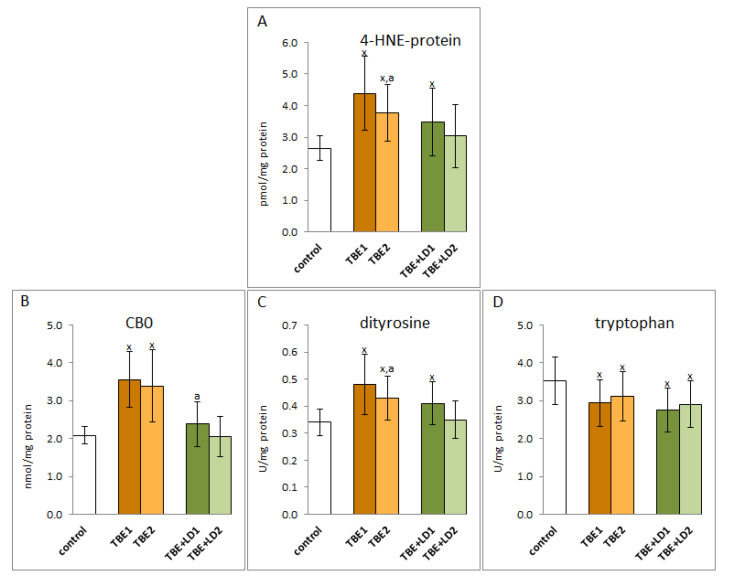
The effect of TBEV infection (TBE1) and TBEV+Bb/Ap co-infection (TBE+LD1) as well as therapy (TBE2 and TBE+LD2) on the protein structure modifications, visible as the levels of 4-HNE-protein adducts (**A**), protein carbonylated groups (CBD) (**B**), protein dityrosine (**C**), and tryptophan (**D**) assessed in plasma of patients compared to healthy subjects (control). The mean values for TBE patients (before–TBE1 and after treatment–TBE2, *n* = 40), patients with TBE+LD (before–TBE+LD1 and after treatment–TBE+LD2, *n* = 6) and control subjects (*n* = 20) with SD are shown as follows: x—differences vs. control group, *p* < 0.05; a—differences vs. TBE1 group, *p* < 0.05.

**Figure 9 pathogens-11-00384-f009:**
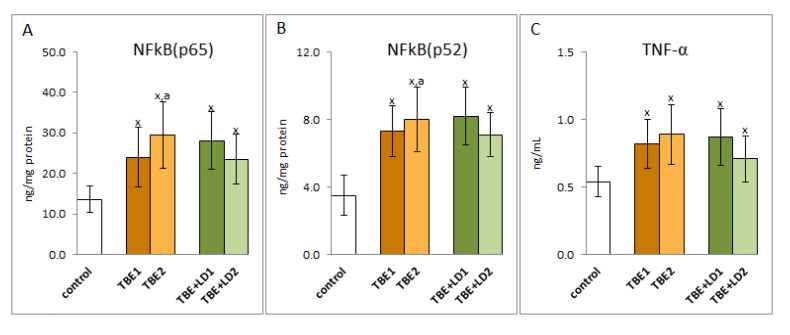
The effect of TBEV infection (TBE1) and TBEV+Bb/Ap co-infection (TBE+LD1) as well as therapy (TBE2 and TBE+LD2) on the effectiveness of the NFkB transcription factor, visible in the expression of subunits NFkB (p65 (**A**) and p52 (**B**)) assessed in granulocytes and product its activity—TNFα (**C**) measured in plasma of patients and compared with healthy subjects (control). The mean values for TBE patients (before–TBE1 and after treatment–TBE2, *n* = 40), patients with TBE+LD (before–TBE+LD1 and after treatment–TBE+LD2, *n* = 6) and control subjects (*n* = 20) with SD are shown as follows: x—differences vs. control group, *p* < 0.05; a—differences vs. TBE1 group, *p* < 0.05.

**Figure 10 pathogens-11-00384-f010:**
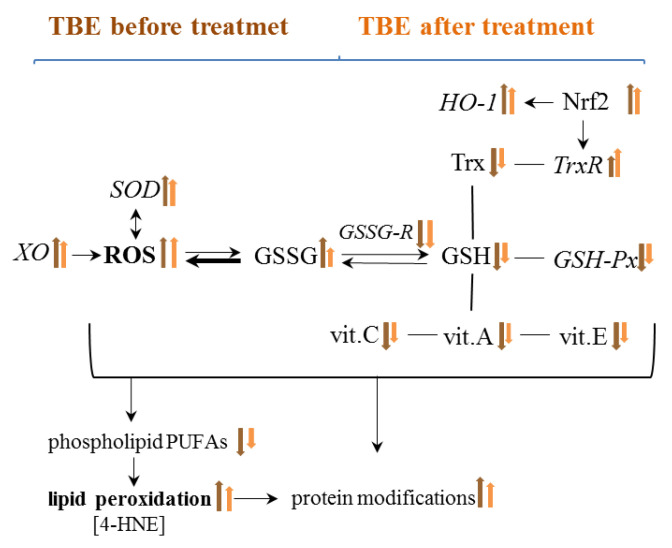
Graphical presentation of the effect of TBE infection on metabolic changes in the blood of patients before and after treatment.

**Table 1 pathogens-11-00384-t001:** Demographic and clinical characteristics of patients with TBEV infection (TBE) and TBEV+Bb/Ap co-infection (TBE+LD/HGA) compared to healthy subjects.

	CTR	TBE	TBE+LD/HGA
Age (Years)	41 (28–55)	40 (23–58)	42 (22–63)
Sex	5/20 female (25%) 15/20 male (75%)	14/40 female (35%) 26/40 male (65%)	4/6 female (67%) 2/6 male (33%)
Confirmed tick bite	0/20 (0%)	21/40 (52%)	5/6 (83%)
Time since tick bite (days)	-	26.8 ± 16.6	17.58 ± 4.95
Duration of hospitalization (days)	-	12.85 ± 2.6	12.34 ± 1.97
Duration of symptoms (days)	-	7.5 ± 6.9	4.26 ± 2.63
Clinical form			
Meningitis	0/20(0%)	25/40 (62.5%)	5/6 (83%)
Meningoencephalitis	0/20 (0%)	15/40 (37.5%)	1/6 (17%)
Meningoencephalomyelitis	0/20 (0%)	0/40 (0%)	0/6 (0%)

**Table 2 pathogens-11-00384-t002:** Comparison of laboratory data of patients with TBEV infection (TBE) and TBEV+Bb/Ap co-infection (TBE+LD/HGA1) as well as therapy (TBE2 and TBE+LD/HGA2) compared to healthy subjects.

	Normal Range	TBE		TBEV+Bb/Ap	
		At Admission	After Recovery	At Admission	After Recovery
*Complete blood count*					
WBC [10^3^/μL]	4.00–10.00	10.16 ± 2.37	6.11 ± 1.48	7.27 ± 1.36	5.37 ± 1.46
Neutrophils [%]	40.0–72.0	73.77 ± 9.52	51.18 ± 7.93	60.33 ± 7.07	42.58 ± 11.58
Lymphocytes [%]	18.00–48.00	16.2 ± 7.76	34.83 ± 6.35	27.37 ± 6.11	42.72 ± 11.24
Monocytes [%]	2.50–10.00	8.99 ± 2.53	9.79 ± 2.15	10.15 ± 2.61	9.56 ± 3.18
RBC [10^6^/μL]	4.00–5.50	4.40 ± 0.43	4.50 ± 0.32	4.28 ± 0.56	4.14 ± 0.62
HGB [g/dL]	12.00–16.00	13.32 ± 1.28	13.68 ± 1.09	12.67 ± 1.26	12.5 ± 1.39
PLT [10^3^/μL]	130–350	251.44 ± 87.43	270.59 ± 134.4	262.67 ± 66.98	218.4 ± 19.42
CRP [mg/L]	0.00–5.00	11.52 ± 15.61	0.93 ± 0.54	2.35 ± 2.08	0.81 ± 0.26
Glucose [mg/dL]	70–110	96.77 ± 10.27	89.62 ± 12.1	92.67 ± 8.5	91.75 ± 5.97
Creatinine [mg/dL]	0.50–0.90	0.88 ± 0.16	0.82 ± 0.12	0.79 ± 0.08	0.76 ± 0.12
ALT [U/I]	0–31	21.10 ± 19.17	24.17 ± 17.20	17.75 ± 11.27	12 ± 5.24
AST [U/I]	0–32	15.59 ± 5.26	22.42 ± 9.23	17 ± 3.67	16.33 ± 3.21
*CSF analysis*					
Cytosis [cells/µL]	0–5	169.7 ± 115.64	26.90 ± 25.59	99 ± 120.57	14.8 ± 12.15
Protein [mg/dL]	15–45	75.95 ± 20.56	54.46 ± 26.32	69 ± 58.49	48.2 ± 27.74

## Data Availability

The data presented in this study are contained within the article.

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
