# Peer review of "Metabolic Response to Tick-Borne Encephalitis Virus Infection and Bacterial Co-Infections"

_pathogens, 2022, doi:10.3390/pathogens11040384_

Round 1
Reviewer 1 Report
This study aimed to compare the metabolic response in patients infected with TBEV /TBEV plus other bacteria, pre and after treatment.
They found that infection promoted the increase of ROS while reduced the antioxidant defense systems (glutathione and thiredoxin systems). The modification of phospholipids containing polyunsatered fatty acids and the increase of lipid peroxidation have been also estimated, and so on.
This study is well conducted and could pave the way for pharmacological strategies.
However, there are some parts to modify to make this work more smooth for the reader. In particular it should be addressed to the discussion section that is very impersonal and the argument of your results appears to be detached from the whole section. It seems that you are only talking about the results obtained by other authors, but you do not correlate them with your own.
Introduction
L 39-49 Please, make explicit all acronyms
Results
In this section has been reported a lot of information that you should move to the discussion section. You have to present only results obtained and limit yourself to this
Discussion
L322-324. Please, rephormulate this sentence.
L326-328 which study? What about your results?
L348-350 are the results you obtained? please, be more specific.
L 353 This mechanism may explain the slight differences in ROS levels ....detected in this study
L 365. It could be releted to
L-362-390 how do you relate these studies to your results?
L 447 as evidenced in this study in which...
L452. in your study?
I think you should proceed like this throughout the manuscript so that it looks like as your research paper and not an excursus based on other studies
Author Response
Thank you very much for the careful analysis of the manuscript and suggested items for improvement. Consequently, we modified the introduction and the results, and re-edited the discussion. We hope that manuscript is more readable in the current version.
At the same time, we would like to inform you that both the introduction, the results and the discussion have been partially redrafted and shortened in order to increase the clarity of the presented data, including the clear specification of the results obtained during the implementation of this work and the literature data. Efforts were also made to eliminate all repetitions.
Responses to individual comments are provided below.
Introduction
Comment: L 39-49 Please, make explicit all acronyms
Answer: Clarification of acronyms has been introduced as follows
TBEV belongs to the Flaviviridae family and, as other viruses from this family is a single-stranded, positive-sense ribonucleic acid (RNA) virus.. It means that its RNA is equivalent of mammalian mRNA and can be directly translated, in this case into a single polyprotein, the processing of which generates 3 structural and 7 nonstructural proteins, including the highly conserved nonstructural glycoprotein NS1, which plays an important role in the replication of its genome, and is typical for all flaviviruses. Additionally, this protein may be responsible for at least some neurotoxic effects, as it is able to cause reactive oxygen species (ROS) generation, although the expression of NF-E2-related factor 2 (NRF2) and antioxidant response element (ARE)-dependent genes is also increased by it. On the other hand, at a later stage of the infection, NS1 is secreted and bound with proteins of the complement system, which inactivates them and leads to an impairment of the immune reaction [3,4].
Results
Comment: In this section has been reported a lot of information that you should move to the discussion section. You have to present only results obtained and limit yourself to this
Answer: As suggested by the Reviewer, the literature considerations, i.e. the entire first paragraph of the Results, was removed. Only a description of the results was left. Please look at the manuscript.
Discussion
Comments: L322-324. Please, rephormulate this sentence.
L326-328 which study? What about your results?
L348-350 are the results you obtained? please, be more specific.
L 353 This mechanism may explain the slight differences in ROS levels ....detected in this study
L 365. It could be related to
L-362-390 how do you relate these studies to your results?
L 447 as evidenced in this study in which...
L452. in your study?
I think you should proceed like this throughout the manuscript so that it looks like as your research paper and not an excursus based on other studies
Answer: As suggested by the Reviewer, the entire discussion has been changed. Therefore, we do not respond to individual comments, but please read the full text. The deliberations on the literature data were reduced and efforts were made to eliminate repetitions, which resulted in a significant change in the structure of the discussion. It seems to us that the discussion is now more compact and clearly identifies elements of the literature data, and we hope that it unequivocally highlights the results of our work. We hope that the introduced modifications have improved the quality of the manuscript in line with the Reviewer's suggestions.
Reviewer 2 Report
In submitted article paper the authors Dobrzynska et al. present results of detailed analysis of oxidative stress (OS) markers and an associated metabolic changes in 40 cases of thick-born encephalitis (TBE) including 6 cases of co-infections with TBE virus and another thick- born bacterial pathogen (Borrelia burgdorferi and Anaplasma phagocytophilum).
They compared levels of OS markers in acute and convalescent phase of the disease (before and after therapy) and look for their differences associated with the bacterial co-infection. As elevated OS markers were recorded in comparison with healthy controls group, especially in the acute phase of infection. Surprisingly, the bacterial co-infection in this respect did not exert significant synergistic effect, which was explained by possible competition between the pathogens involved.
The paper shows a lot of data from biochemical analyses which in TBE were not studied in such a detail so far. The methods used were mostly well described and the results were well interpreted in relation to lipid metabolism , enzymatic changes involved in redox homeostasis and in the pathogen defence. However, in the context of infection accompanied with significant inflammatory reaction, these results could be expected and their significance is rather moderate. Moreover, the co-infection group is too small and not well characterised to serve as representative for identification of the co-infection biomarkers.
Specific comments:
Both the Introduction and the Discussion are too long. Some data which are generally known (i.e. lines 34-35, 93-96, 99-100, 319-322,341-342…), or described in the detail elsewhere may be omitted or cited, respectively. Some statements are repeated in both the Paragraphs.
Graphical illustration of the association between the markers studied would be helpful.
Materials and methods:
Co-infection group should be better characterized: Infection was associated with Borrelia burgdorferi sensu stricto or sensu lato? How many patients with LB or HGA? How many patients was diagnosed by exanthema migrans or neuroborreliosis? Time- relation of bacterial infection diagnoses to the diagnosis of TBE (with respect to differences in the disease duration and incubation periods) should be described. Detection of ROS is described only in unfractionated blood, not in granulocytes and lymphocytes.
Results: When considered their therapeutic potential, correlation between the levels of OS markers and other markers of inflammation, especially those routinely measured in CSF should be studied.
Author Response
Thank you very much for the careful analysis of the manuscript and suggested items for improvement. Consequently, we modified the introduction and the results, and re-edited the discussion. We hope that manuscript is more readable in the current version.
At the same time, we would like to inform you that both the introduction, the results and the discussion have been partially redrafted and shortened in order to increase the clarity of the presented data, including the clear specification of the results obtained during the implementation of this work and the literature data. Efforts were also made to eliminate all repetitions.
Responses to individual comments are provided below.
Comment:
Materials and methods:
Co-infection group should be better characterized: Infection was associated with Borrelia burgdorferi sensu stricto or sensu lato? How many patients with LB or HGA? How many patients was diagnosed by exanthema migrans or neuroborreliosis? Time- relation of bacterial infection diagnoses to the diagnosis of TBE (with respect to differences in the disease duration and incubation periods) should be described. Detection of ROS is described only in unfractionated blood, not in granulocytes and lymphocytes.
Answer:
1/ The clinical description of the group of patients has been completed as follows:
“Co-infection was diagnosed if one patient was infected with at least two different pathogens. LD was defined on the basis of clinical presentation of erythema migrans (1 patient) or fulfilled criteria for neuroborreliosis (1 patient). Moreover, in two patients anti-Borrelia burgdorferi IgM and IgG antibodies were detected and the patients were treated for probable case of NB. Infection was associated with Borrelia burgdorferi sensu lato. In four cases human granulocytic anaplasmosis based on clinical picture, laboratory tests and PCR was diagnosed. In TBE patients time since tick bite to symptoms onset was 26.8±16.66 days, while in patients co-infected with A. phagocytophilum or B. burgdorferi it was 17.58±4.95 days. “
The information about time since time bite to the onset of symptoms was also given in Table 1.
2/ In fact, the methodology does not take into account all the data on the obtaining granulocytes and lymphocytes from blood and the determination of ROS in these cells. We have completed the methodology as follows:
“Blood samples were collected into ethylenediaminetetraacetic acid (EDTA) tubes and a two-stage centrifugation was carried out. In the first stage, the sample was centrifuged at 3,000×g (4°C) to separate the plasma, buffy coat and erythrocytes. To obtain clear fraction of granulocytes, obtained buffy coat were layered on Gradisol G (Aqua-Med ZPAM–KOLASA, Łódź, Poland) and subjected to 25 min centrifugation at 300×g at room temperature. To obtain clear fraction of lymphocytes whole blood was layered on Gradisol L (Aqua-Med ZPAM–KOLASA, Łódź, Poland) and centrifuged for 25 min at 300 g. The lymphocyte containing fraction was collected from above the gradisol layer, washed tree times in PBS (3 min centrifugation at 300g) and resuspended in PBS containing a protease inhibitor mix. Cells in the middle fraction were collected, washed, and re-suspended in PBS containing a proteasome inhibitor mix. The purity of the obtained cell fraction was examined microscopically (Nikon Eclipse Ti, Nikon Instruments Inc., NY, USA).”
“50 μL of blood and granulocytes/lymphocytes obtained from 2 ml of blood, collected in heparinized capillary tubes were analyzed. Among spin trapping (otherwise labelled probe) molecules, suitable for biological utilization, 1-hydroxy-3-methoxycarbonyl-2,2,5,5-tetramethylpyrrolidine (CMH, Noxygen Science Transfer & Diagnostics, Germany) was adopted. A 400 µM CMH solution was prepared in buffer (Krebs-Hepes buffer (KHB) containing 25 μM deferroxamine methane-sulfonate salt (DF) chelating agent and 5 μM sodium diethyldithio-carbamate trihydrate (DETC) at pH 7.4. Blood and granulocytes/lymphocytes in PBS was immediately treated with CMH (1 : 1) [69]…”
Comment:
Results: When considered their therapeutic potential, correlation between the levels of OS markers and other markers of inflammation, especially those routinely measured in CSF should be studied.
Answer:
Correlations between markers of oxidative stress and routinely measured markers of inflammation were assessed and relevant information was included in the methodology and in the results section
Methodology
“Correlation between the levels of oxidative stress markers and other markers of inflammation was calculated by use of Spearman rank correlation test”
Results
“The statistical analysis performed in TBE group before treatment, showed negative correlation between SOD and pleocytosis in CSF (R=-0.35), negative correlation of NPs with WBC in blood (R=-0.38) and positive correlation of HNE-b with CRP, pleocytosis and protein concentration in CSF (R=0.38; 0.44, 0.32, respectively). After treatment only negative correlation of Vitamin C with protein concentration in CSF (R=-0.35) was observed.”
Comment:
Graphical illustration of the association between the markers studied would be helpful
Answer:
Figure 7 presenting the obtained-representative results of the research was added into the discussion.